# Biosensor-Driven IoT Wearables for Accurate Body Motion Tracking and Localization

**DOI:** 10.3390/s24103032

**Published:** 2024-05-10

**Authors:** Nouf Abdullah Almujally, Danyal Khan, Naif Al Mudawi, Mohammed Alonazi, Abdulwahab Alazeb, Asaad Algarni, Ahmad Jalal, Hui Liu

**Affiliations:** 1Department of Information Systems, College of Computer and Information Sciences, Princess Nourah bint Abdulrahman University, P.O. Box 84428, Riyadh 11671, Saudi Arabia; naalmujaly@pnu.edu.sa; 2Faculty of Computing ad AI, Air University, E-9, Islamabad 44000, Pakistan; 211651@students.au.edu.pk; 3Department of Computer Science, College of Computer Science and Information System, Najran University, Najran 55461, Saudi Arabia; nalmdwi@nu.edu.sa (N.A.M.); afalzeb@nu.edu.sa (A.A.); 4Department of Information Systems, College of Computer Engineering and Sciences, Prince Sattam bin Abdulaziz University, Al-Kharj 16273, Saudi Arabia; mn.alonazi@psau.edu.sa; 5Department of Computer Sciences, Faculty of Computing and Information Technology, Northern Border University, Rafha 91911, Saudi Arabia; asad.algani@nbu.edu.sa; 6Cognitive Systems Lab, University of Bremen, 28359 Bremen, Germany

**Keywords:** machine learning, segmentation, feature fusion, multi-layer perceptron, Yeo–Johnson

## Abstract

The domain of human locomotion identification through smartphone sensors is witnessing rapid expansion within the realm of research. This domain boasts significant potential across various sectors, including healthcare, sports, security systems, home automation, and real-time location tracking. Despite the considerable volume of existing research, the greater portion of it has primarily concentrated on locomotion activities. Comparatively less emphasis has been placed on the recognition of human localization patterns. In the current study, we introduce a system by facilitating the recognition of both human physical and location-based patterns. This system utilizes the capabilities of smartphone sensors to achieve its objectives. Our goal is to develop a system that can accurately identify different human physical and localization activities, such as walking, running, jumping, indoor, and outdoor activities. To achieve this, we perform preprocessing on the raw sensor data using a Butterworth filter for inertial sensors and a Median Filter for Global Positioning System (GPS) and then applying Hamming windowing techniques to segment the filtered data. We then extract features from the raw inertial and GPS sensors and select relevant features using the variance threshold feature selection method. The extrasensory dataset exhibits an imbalanced number of samples for certain activities. To address this issue, the permutation-based data augmentation technique is employed. The augmented features are optimized using the Yeo–Johnson power transformation algorithm before being sent to a multi-layer perceptron for classification. We evaluate our system using the K-fold cross-validation technique. The datasets used in this study are the Extrasensory and Sussex Huawei Locomotion (SHL), which contain both physical and localization activities. Our experiments demonstrate that our system achieves high accuracy with 96% and 94% over Extrasensory and SHL in physical activities and 94% and 91% over Extrasensory and SHL in the location-based activities, outperforming previous state-of-the-art methods in recognizing both types of activities.

## 1. Introduction

Human locomotion activity recognition is a rapidly emerging field that analyzes and classifies many types of physical activity and locomotion activity using smartphone sensors [1]. Smartphones are an ideal platform for this research because they are widely available, equipped with a variety of sensors [2], and are commonly carried by individuals. In modern smartphones, a rich array of sensors including accelerometers, gyroscopes, magnetometers, GPS, light sensors, barometers, and microphones, are utilized for comprehensive human locomotion activity recognition. These sensors allow for the accurate detection and analysis of user movements and environmental interactions, enhancing the performance of activity recognition systems. Human activity recognition using smartphone sensors has a wide range of applications in various fields. Some of the most common applications include fitness and health monitoring, and elderly care (to monitor the movements of elderly individuals and detect any signs of falls or other accidents [3]. This can provide peace of mind for caregivers and family members and can also be used to trigger an alert if assistance is needed), transportation, environmental monitoring, sports, and marketing (to track and analyze the movements of individuals in different retail environments. This information can be used to better understand consumer behavior and make decisions about product placement and advertising), safety, and security [4].

Researchers have used different machine-learning approaches to recognize human activity through smartphone sensors. These approaches have several advantages, including the ability to recognize complex patterns and classify multiple activities simultaneously. Additionally, machine learning algorithms can improve over time as they are fed more data, making them more accurate and reliable [5,6]. However, there are also some disadvantages to using machine learning for activity recognition. One of the main disadvantages is the requirement for large amounts of labeled data to train the algorithms effectively. This can be time-consuming and expensive, especially when dealing with numerous activities or users. Another potential issue is the sensitivity of the algorithms to changes in sensor placement or sensor data quality. If the sensor is not positioned correctly or the data are noisy, the accuracy of the machine-learning algorithm can be significantly reduced [7].

The field of HAR is impeded by a variety of challenges, including sensor heterogeneity across devices [8], which complicates the development of universal application. Different smartphone models are equipped with various types of sensors that have differing specifications and capabilities. This variation complicates the development of universal applications that perform consistently across all devices. Another big challenge researchers face is that of noise in the raw sensor data [9]. Smartphone sensors frequently capture data contaminated with noise, which can significantly affect accuracy. The precision of sensors varies widely between devices, often depending on the hardware quality and sensor calibration. Additionally, variations in sensor sampling frequencies [10], mean that the rate at which sensor sample data can fluctuate is influenced by other processes running on the device. This inconsistency can lead to challenges in capturing the real-time, high-resolution data necessary for precise motion recognition. Similarly, sensor characteristics can change over time due to aging hardware or software updates, leading to a data drift. This phenomenon can degrade the performance of motion recognition algorithms that were trained on data from newer or different sensors. Continuous sensor data collection is resource-intensive, consuming significant battery life and processing power. Managing these resources efficiently while maintaining accurate motion detection is a major challenge. Moreover, different users may carry their smartphones in various positions (e.g., pocket, hand, or bag) [11], leading to vastly different data profiles. Algorithms must be robust enough to handle these variations to ensure accurate motion recognition. Moving forward, another challenge is the privacy of the user [12], collecting and analyzing sensor data increases privacy concerns, as such data can inadvertently reveal sensitive information about a user’s location and activities. Ensuring data privacy and security while collecting and processing sensor data is critical. Lastly, the complexity of human activities and the limited spatial coverage of sensors add to the difficulty of capturing a comprehensive range of motions, highlighting the multifaceted nature of these technological hurdles. In this study, we developed a system that recognizes human movements along with location, and ultimately, can provide valuable insights into an individual’s physical and localization activity levels. We processed the raw sensor data for physical and location-based activity separately. In the first stage, we denoised the data using the Butterworth filter [13] and Median filter [14]. In the second stage, we segmented the long sequence signal data into small pieces using the Hamming windowing technique [15]. After that, features were extracted. The Variance threshold feature selection method [16,17] is used for feature selection, the selected feature vectors are balanced using the data augmentation technique, and after that, the augmented data are well optimized before classification using the Yeo–Johnson power transformation technique. Finally, physical and localization activity classification is performed by a multi-layer perceptron (MLP). The contribution of this research is described below:Implemented separate denoising filters for inertial and GPS sensors, significantly enhancing data cleanliness and accuracy.Developed a robust methodology for concurrent feature extraction from human locomotion and localization data, improving processing efficiency and reliability.Established dedicated processing streams for localization and locomotion activities, allowing for more precise activity recognition by reducing computational interference.Applied a novel data augmentation technique to substantially increase the dataset size of activity samples, enhancing the robustness and generalizability of the recognition algorithms.Utilized an advanced feature optimization algorithm to adjust the feature vector distribution towards normality, significantly improving the accuracy of activity recognition.

The research has been divided into the following sections: Section 2 discusses some literature review in the field of human activity recognition, and then material and methods, including noise removal, signal windowing and segmentation, feature extraction, feature selection, and optimization presented in Section 3. Section 5 analyzes the computational complexity of the proposed system, Section 6 presents a discussion and limitations. Finally, in Section 7, the research study is concluded.

## 2. Related Work

Scientists have explored different approaches to studying how to analyze human motion, both inside and outside [18]. These approaches can be divided into two main groups: ones that depend on motion-based analysis and ones that depend on vision-based analysis [19,20], whilst some are based on sensors-based methods that make use of sensors such as accelerometers, gyroscopes, GPS, light, mic, magnetometers, mechanomyography, ECG (electrocardiogram), EMG (electromyogram), and geomagnetic sensors. Vision-based methods, on the other hand, use cameras such as Microsoft Kinect [21], Intel Real sense, video cameras, and dynamic vision sensors [22]. 

The related work by Hsu et al. [23] involved a method of human activity recognition that utilized a pair of wearable inertial sensors. On the subject’s wrist, one sensor was mounted, while the other was mounted on the ankle. The collected sensor data, including accelerations and angular velocities, was wirelessly sent to a central computer for processing. Using a nonparametric weighted feature extraction algorithm and principal component analysis, the system could differentiate between various activities. While the method boasted the high points of portability and wireless data transmission, it was limited by the use of only two sensors, potentially missing out on capturing the full spectrum of human movement and requiring a dependable wireless connection to function effectively. To improve upon this, the proposed solution in the research includes the deployment of additional sensors on various body parts such as the torso, backpack, hand, and pocket to provide a more comprehensive capture of human motion. Additionally, integrating sensors embedded within smartphones eliminates the need for a continuous wireless connection, facilitating the recognition of human activities and locations with enhanced reliability and context awareness. In the research by A-Basset et al. [24], a novel approach to human activity recognition is introduced where sensor data is treated as visual information. Their method considers human activity recognition as analogous to image classification, converting sensor data into an RGB image format for processing. This enables the use of multiscale hierarchical feature extraction and a channel-wise attention mechanism to classify activities. The strength of this system lies in its innovative interpretation of sensor data, which allows for the application of image classification techniques. However, its reliance on small datasets for training raises questions about how well it can be generalized to real-world situations. The uncertainty regarding the computational and space complexity also poses concerns about the system’s scalability. The proposed enhancement of this system involves training on larger and more diverse datasets to enhance the system’s capacity to generalize across various scenarios. By ensuring that the system is robust when handling larger datasets, the solution seeks to maintain computational efficiency while scaling up to more complex applications. Konak et al.’s [25] method for evaluating human activity recognition performance employs accelerometer data, which is categorized into three distinct classes based on motion, orientation, and rotation. The system utilizes these categories either individually or in combination to assess activity recognition performance and employs a variety of classification techniques, such as decision trees, naive Bayes, and random forests. The primary limitation of this method is its training on a dataset derived from only 10 subjects, raising concerns about its generalizability to a broader population. Additionally, the study relies on common machine learning classifiers, which may not be as effective as more advanced models. In contrast, the proposed model in the research under discussion utilizes the Extrasensory and Huawei dataset, which includes data from more subjects, thus providing a more robust and generalizable system that achieves state-of-the-art performance. The research by Chetty et al. [26] presents an innovative data analytic method for human activity recognition using smartphone inertial sensors, utilizing machine learning classifiers like random forests, ensemble learning, and lazy learning. The system distinguishes itself through its feature ranking process informed by information theory, which optimizes for the most relevant features in activity recognition. Despite the innovative approach, the system’s reliance on a single dataset for training is its primary limitation. This constraint could hinder the model’s ability to generalize to unobserved scenarios and potentially lead to degraded performance in real-world applications. The proposed solution to these limitations involves a system trained on two benchmark datasets that encompass a wider variety of activities. It includes the Extrasensory dataset, which is notable for being collected in uncontrolled, real-world environments without restrictions on participant behavior. This approach is intended to enhance the system’s reliability and applicability to a broader range of real-life situations, thereby making it a more robust solution for activity recognition.

The study by Ehtisham-ul-Haq et al. [27] introduced an innovative context recognition framework that interprets human activity by leveraging physical activity recognition (PAR) and learning patterns from various behavioral scenarios. Their system correlated fourteen different behaviors, including phone positions with five daily living activities, using random forest and other machine learning classifiers for performance evaluation. The high points of this method are its use of human activity recognition to infer context and its integration of additional information such as the subject’s location and secondary activities. Nonetheless, the system’s primary reliance on accelerometer data makes it less adept at complex activities, and it lacks more comprehensive data sources like GPS and microphone inputs for enhanced location estimation. The proposed enhancement to this framework includes a more integrated sensor approach, utilizing not only the smartphone’s accelerometer, magnetometer, and gyroscope, but also the smartwatch’s accelerometer and compass, along with smartphone GPS and microphone data. This integration promises increased robustness and accuracy in activity recognition and localization.

### 2.1. Activity Recognition Using Inertial Sensors

Smartphone sensors are more popular for activity recognition. By using these sensors, human activity can be easily detected. The most significant feature of the smartphone is its portability, which means it can be carried easily in any place. In [28], different supervised machine learning algorithms were used to classify human activity. The classification precision was tested using 5-fold cross-validation. They achieved a good accuracy rate for all classifiers. Ref. [29] presented trends in human activity recognition. The survey discussed different solutions proposed for each phase in human activity classification, i.e., preprocessing, windowing and segmentation, feature extraction, and classification. All the solutions are analyzed, and the weaknesses and strengths are described. The paper also presented how to evaluate the quality of a classifier. In [30], a new method was proposed for recognizing human activity with multi-class SVM using integer parameters. The method used in the research consumes less memory, processor time, and power consumption. The authors in [31] analyzed the performance of the two classifiers, that is, KNN and clustered KNN. The classifiers were evaluated with an online activity recognition system using the Android operating system. The system supports online training and classification by collecting data from one sensor called an accelerometer. They started with KNN and then clustered it. The main rationale for utilizing clustered was to reduce the computational complexity of KNN using clusters. The major goal of the article was to examine the performance of algorithms on the phone with limited training data and memory.

### 2.2. Activity Recognition Using Computer Vision and Image Processing Techniques

As previously noted, identifying human activity via smartphone is a convenient method because the smartphone is a portable device that can be readily carried anywhere. The use of an RGB camera for activity recognition [32,33] has some limits and constraints. To monitor a person’s activity with a camera, for example, the individual must be within range of the camera’s eye. Nighttime (changing lighting conditions) is the second most prevalent challenge that researchers face while tracking human activities through a webcam. However, advancements in multimedia tools have mitigated these issues to some extent. To recognize human movement from 2D/3D films and photos, many computer-vision and image-processing algorithms have been utilized [34,35]. Researchers can recognize human activities more easily by employing techniques such as segmentation [36], filtering [37], saliency map detection [38], skeleton extraction [39], and so on. The work described in [40] investigated human activity recognition using a depth camera. The camera first acquired the skeleton data, and then several spatial and temporal features were retrieved. The CNN (Convolutional Neural Network) algorithm was employed to classify the activities. The issue with highlighting an in-depth camera is that noise can occur, leading to misclassification.

## 3. Proposed System Methodology

Data were collected from various raw sensors. The data were denoised in the first step using the Butterworth filter [40]. The Hamming windowing and segmentation approach [41] is then applied. During the third step, we worked with the data from the inertial and GPS sensors. We picked out various features for each of them. To determine the significance of features, we employed the Variance Threshold for feature selection. We noted that certain activities in the Extrasensory dataset had a limited number of samples. To address this issue, we applied data augmentation and subsequently optimized using the Yeo–Johnson power transformation technique before conducting activity recognition. Finally, the activity recognition was performed by the MLP. The flow diagram of the suggested human physical and localization activity model is shown in Figure 1. 

### 3.1. Signal Denoising

There is a risk of noise during data collection. Noise is the undesirable portion of data that we do not need to process. Unwanted data processing lengthens and complicates model training. It also reduces the learning model’s performance. Therefore, noise removal is crucial in data preprocessing. For this reason, we employed a noise-removal filter.

To get rid of the unwanted disruptions that can happen when collecting data, we applied a low-pass Butterworth filter [42,43,44,45] to the inertial sensors. This filter is used in signal processing and aims to make the frequency response as even as possible in the part where it passes signals through. That is why it is called the maximally flat magnitude filter. Equation (1) depicts the general frequency response of the Butterworth filter.
(1)$$H\left(jw\right)=\frac{1}{\sqrt{1+{\left(\frac{\omega }{\omega_{c}}\right)}^{2n}}}$$
where *n* is the order of the filter, *ω* is the passband frequency (also known as the operational frequency), *ω_c_* is the filter’s cut-off frequency, and *j* is the imaginary unit, used to denote the complex frequency. In Figure 2, the original vs. filtered signal for the inertial sensor is shown. Similarly, for processing our GPS data and to enhance its clarity, we used the median filter [46,47,48,49] a robust nonlinear digital filtering technique. The median filter operates by moving a sliding window across each data point in our GPS sequence. Within this window, the data values are arranged in ascending order. The central value, or median, of this sorted list, is then used to replace the current data point. Mathematically, for a given signal, S, and a window of size *n*, at each point *xi* in the signal, we consider:(2)W={(xi − (n−1))/2−(xi−(n−1))/(2+1),….(xi +(n−1))/(2−1)+(xi+(n−1))/2}

The median of this set *W* becomes the new value at *xi* in the filtered signal. In our experiment on the GPS data, we selected a window size of 3. The selected window size ensures that the filter assesses each data point while considering itself and one neighboring point on either side. This particular size strikes a balance by being large enough to effectively suppress noise, and yet, be compact enough to preserve important details and transitions in the GPS data. But it is important to note that there was not enough noise in the GPS signal as inertial sensors.

### 3.2. Signal Windowing and Segmentation

Segmentation is an important concept used in signal processing. The concept of windowing and segmentation [50,51,52,53,54] involves dividing signals into smaller windows instead of processing complete or long sequences. The advantage of windowing is that it allows for easier data processing, reducing complexity and processing time. This makes it more manageable for machine learning or deep learning models to process. We turned to the Hamming windows technique to modulate the signal. Hamming windows, known for their capacity to reduce spectral leakage [55] during frequency domain operations like the Fourier Transform, effectively tackle the side effects that often arise during such analyses. The principle behind the Hamming window is a simple point-wise multiplication of the signal with the window function, which curtails the signal values at both the start and end of a segment. This modulation ensures a minimized side lobe in the frequency response, which is crucial for accurate spectral analyses. 

Mathematically, the Hamming window is represented in Equation (3).
(3)$$\;\;W\left(n\right)=0.54-0.46cos\left(\frac{2\pi n}{N-1}\right)$$
where *w*(*n*) represents the window function, *N* signifies the total points in the window, and *n* spans from 0 to *N* − 1. We utilized a window size of 5 s [56,57]. After generating the Hamming window values based on the aforementioned formula, we multiplied each point in our data segments with its corresponding Hamming value. In Figure 3, we visualized the results through distinct line plots, with each of the five windows represented in a unique color.

### 3.3. Feature Extraction

In this section, we listed all the feature lists used in the study specifically aligned with each type of sensor data. We extract separate features for the physical and localization activities. The subsequent section presents each section comprehensively.

#### 3.3.1. Feature Extraction for Physical Activity

For physical recognition, we processed data from three sensors: magnetometer, gyroscope, and accelerometer [58,59,60,61]. Various statistical features were extracted.

##### Shannon Entropy

Shannon entropy is first extracted, as seen in Figure 4. Shannon entropy [62,63] measures the unpredictability [64,65,66] or randomness of a signal. Mathematically, it can be calculated as:(4)$$H\left(P\right)=-\sum_{i}p_{i}{log}_{2}\left(p_{i}\right)$$
where *p_i_* represents the probability of occurrence of the different outcomes. 

##### Linear Prediction Cepstral Coefficients (LPCCs)

The extraction of LPCCs from accelerometer signals [67], the primary step involves the application of linear predictive analysis (LPA). Given *s*(*n*) as the accelerometer signal, it can be modeled by the relation
(5)$$\;s\left(n\right)=\sum_{k=1}^{p}a_{k}s\left(n-k\right)+e\left(n\right)$$
where *p* represents the order of the linear prediction, *a_k_* are the linear prediction coefficients, and *e*(*n*) denotes the prediction error. The linear prediction coefficients, *a_k_*, derived by minimizing the prediction’s mean square error, were commonly achieved using the Levinson–Durbin algorithm. After obtaining these coefficients, the transition to cepstral coefficients begins. This conversion entails taking the inverse Fourier transform of the logarithm of the signal’s power spectrum. Specifically, the cepstral coefficients are determined through a recurrence relation, where the initial coefficient is the logarithm of the zeroth linear prediction coefficient, and subsequent coefficients are derived using the linear prediction coefficients and previous cepstral coefficients. The LPCCs calculated for different activities can be seen in Figure 4.

##### Skewness

In the context of signal processing for accelerometer data, skewness is a crucial statistical measure that captures the asymmetry [68,69,70] of the signal distribution. To compute the skewness of an accelerometer signal *s*(*n*), where n represents the discrete time index, we first calculate the mean (*μ*) and standard deviation (*σ*) of the signal. Following this, the skewness (*S*) is obtained using the formula.
(6)$$S=\frac{1}{X}\sum_{x=1}^{X}{\left(\frac{s\left(x\right)-\mu }{\sigma }\right)}^{3}\;$$

Here, *X* is the total number of data points in the signal. The formula essentially quantifies the degree to which the signal’s distribution deviates from a normal distribution. A skewness value of zero signifies a symmetric distribution. Positive skewness indicates a distribution with an asymmetric tail extending towards more positive values, while negative skewness indicates a tail extending towards more negative values. Computing the skewness of an accelerometer signal can provide insights into the distribution characteristics of the signal. Figure 5 show the skewness for different locomotion activities.

##### Kurtosis

Kurtosis is a statistical measure used to describe the distribution of observed data around the mean. Specifically, it quantifies the probability distribution of a real-valued random variable. In the context of signals, kurtosis [71] can be particularly informative as it can capture the sharpness of the distribution’s peak and the heaviness of its tails. This, in turn, can indicate the occurrence of abrupt or high-magnitude changes in the acceleration data, which may be characteristic of specific activities or movements. The formula for kurtosis is given by:(7)$$\mathrm{Kurtosis}\;\left(x\right)=F\left[{\left(\frac{x-\mu }{\sigma }\right)}^{4}\right]-3$$
where *F* denotes the expected value, *μ* is the mean, and *σ* is the standard deviation. A kurtosis value greater than zero indicates that the distribution has heavier tails and a sharper peak compared to a normal distribution. Conversely, a value that is less than zero suggests that the distribution has lighter tails. In our analysis, we extracted the kurtosis from the accelerometer signals corresponding to different activities. This enabled us to discern and distinguish the nature of signal distributions for activities such as cooking, sitting, or cleaning. For instance, a sudden or vigorous activity might exhibit a distribution with a higher kurtosis value, indicating rapid changes in acceleration, whereas more steady or uniform activities might have a lower kurtosis value. In Figure 6, the kurtosis plot is presented. 

#### 3.3.2. Feature Extraction for Localization Activity

For localization activities, we try to capture the complicated movement patterns by extracting a set of distinct features. We extracted the Total Distance, Average Speed, Maximum Displacement, Direction Change features, heading angles, skewness, kurtosis, step detection, and MFCCs [72].

##### Mel-Frequency Cepstral Coefficients (MFCCs)

In human localization using audio signals, MFCCs [73] play a pivotal role in determining the direction, proximity, and potential movement patterns. We begin with the pre-emphasis of the signal *s*(*n*), accentuate its high frequencies, a step mathematically represented as:(8)$$\overset{´}{s}=s\left(n\right)-\alpha \times s\left(n-1\right)$$
where *α* is commonly set to 0.97. This amplification aids in emphasizing the subtle changes in audio signals that may result from human movement or orientation changes. The signal is then split into overlapping frames to analyze temporal variations, and each frame is windowed, often using the Hamming window, to mitigate spectral leakage. The short-time Fourier transform (STFT) offers a frequency domain representation of each frame, and its squared magnitude delivers the power spectrum. As human auditory perception is nonlinear, this spectrum is translated to the Mel scale using triangular filters. This transformation is governed by:(9)$$m=2595\times {log}_{10}\left(1+\frac{f}{700}\right)$$

The mathematics above ensures that the extracted features align with human auditory perception. The logarithm of this Mel spectrum undergoes the Discrete Cosine Transform (DCT), producing the MFCCs. By retaining only the initial coefficients, one captures the essential spectral shape, pivotal for discerning sound characteristics that aid in human localization. The MFCCs calculated for localization activities can be seen in Figure 7.

##### Step Detection

To understand the steps [74,75,76,77,78] from accelerometer data, we harness the magnitude of the acceleration vector. This magnitude is essentially a scalar representation of the combined accelerations in the *x*, *y*, and *z* axes. Mathematically, given the acceleration values *a_x_*, *a_y_*, *a_z_* in the respective axes, the magnitude *M* is calculated using the formula:(10)$$M=\sqrt{a_{x}^{2}+a_{y}^{2}+a_{z}^{2}}$$

Once we have the magnitude of acceleration, the periodic nature of indoor or outdoor environments produces recognizable peaks in this signal. Each peak can correspond to a step, and by detecting these peaks, we can estimate the number of steps taken. The peak detection is anchored on identifying local maxima in the magnitude signal that stand out from their surroundings. The step detected [79] for indoor and outdoor activities can be seen in Figure 8.

##### Heading Angle

The heading angle [80,81] plays a pivotal role in determining the orientation or direction a person is facing. As humans navigate through environments, whether they are indoor spaces like shopping malls or outdoor terrains like city streets, understanding their heading is crucial for applications ranging from pedestrian navigation systems to augmented reality. The heading angle, often termed the azimuth, denotes the angle between the North direction (assuming a geomagnetic North) and the projection of the magnetometer’s reading onto the ground plane. Mathematically, the heading angle *θ* can be calculated using the magnetic field components *A* and *B* as:(11)θ=arctan2(B,A)
where *arctan*2 is the two-argument arctangent function, ensuring the angle [82] lies in the correct quadrant and providing a result in the range [−180, 180]. The heading for indoor and outdoor activity can be seen in Figure 9.

### 3.4. Feature Selection Using Variance Threshold

In this experiment, we applied the variance threshold method [83,84,85,86] to the feature vector. The goal was to identify and retain only those features that showed significant variation across all features, ensuring that our dataset was as informative as possible. The feature mean, standard deviation, and total distance, exhibited a relatively low variance and were therefore removed, while all other features were retained. Variance Threshold is a simple filter-based feature selection method. It removes all features whose variance across all samples does not meet a specific threshold. The rationale behind this approach is straightforward: features that do not vary much within themselves are less likely to be informative. The variance of a feature *X* is given by:(12)$$Var\left(x\right)\;=\;\frac{1}{n}\Sigma_{i=1}^{n}{\left(xi-\bar{x}\right)}^{2}$$
where: *n* is the number of samples; *xi* is the value of the feature *X* for the *i*th sample; and $\bar{x}$ is the mean value of the feature *X* across all samples. In the context of the variance threshold method, we compare the variance of each feature against a pre-defined threshold. Features with variances below this threshold are considered non-informative and are removed. The algorithm working of the variance threshold is shown in Algorithm 1.
**Algorithm 1:** Variance Threshold Feature Selection**1: Input**: Dataset *D* with *m* features: *f*1, *f*2, …, *fm*. 
*Variance threshold value τ.*
 k: Desired number of features to select. 
**2: Output**:  A subset of features whose variance is above 
**3: Initialization:**
 Create an empty list *R* to store the retained features 
**4: Feature Selection:**
   **For** each feature *f_i_* in (*D*). Compute the variance *v_i_* of *f_i_*
    Add *f_i_* to the list *R*
    **end for**

**5: Return:**
 Return the list *R* as the subset of features with variance above *τ*. 
**6: End**

### 3.5. Feature Optimization via Yeo–Johnson Power Transformation

We perform feature optimization before moving on to classification. In simple terms, feature optimization makes the feature clearer to the model. We opted to optimize the specified feature vector after selecting relevant features for the model using the Variance Threshold. For this purpose, we utilized the Yeo–Johnson power transformation method. The Yeo–Johnson power transformation [87] is a statistical method used to transform non-normally distributed data into a normal or Gaussian-like distribution. This method is highly valuable in machine learning, as many algorithms assume that the data follow a normal distribution. By transforming the data using the Yeo–Johnson method, we can enhance the performance of these algorithms and make the results more reliable. The method uses a power transformation to map the original data into a new distribution, with the power being a parameter that is estimated from the data. Mathematically, the Yeo–Johnson optimization is given in Equation (13).
(13)$$\psi \left(x\right)=\{\begin{array}{c} \frac{{\left(x+1\right)}^{\lambda }-1}{\lambda }\;\;\;\;\;\;\lambda \neq 0\;and\;x\geq 0,\\ ln\left(x+1\right),\lambda =0\;and\;x\geq 0,\\ \frac{-{\left(-x+1\right)}^{2-\lambda }-1)}{2-\lambda },\;\lambda \neq 2\;and\;x<0,\\ ln\left(-x+1\right),\;\;\lambda =2\;and\;x<0 \end{array}$$

### 3.6. Data Augmentation

In addressing the challenge of class imbalance in datasets, the permutation technique [88,89,90] emerges as a novel data augmentation method, which is particularly effective for sequential or time-series data. At its core, the permutation technique involves dividing a signal into multiple non-overlapping segments and then rearranging these segments in various orders to generate new samples. For example, given a time-series signal divided into three segments, A, B, and C, permutations can produce sequences such as B–A–C, C–B–A, or even B–C–A. This method capitalizes on the inherent structure and patterns within the data, creating diverse samples that maintain the original signal’s fundamental characteristics. When applied to the minority class in an imbalanced dataset, the permutation technique can artificially expand the number of samples, thus bridging the gap between the majority and minority classes. This ensures that the learning algorithm is exposed to a broader spectrum of data variations from the minority class, potentially enhancing its ability to generalize and reducing the bias towards the majority class. 

### 3.7. Proposed Multi-Layer Perceptron Architecture

Our proposed MLP architecture [91,92,93,94] was designed to handle the complexity and variability inherent in the sensor data collected. With the manual feature extraction and subsequent optimization processes we employed, our MLP [95,96,97,98] was strategically tasked with classifying a refined feature vector that encapsulates essential information for robust activity recognition.

#### 3.7.1. Architecture Overview

Input Layer: The size of the input layer directly corresponds to the number of features extracted and optimized from the sensor data. In our study, the dimensionality of the input layer was adjusted based on the dataset being processed, aligning it with the feature vector size derived after optimization.Hidden Layers: We include three hidden layers. The first and second hidden layers are each composed of 64 neurons, while the third hidden layer contains 32 neurons. We utilized the ReLU (rectified linear unit) activation function across these layers to introduce necessary nonlinearity into the model, which is crucial for learning the complex patterns present in the activity data.Output Layer: The size of the output layer varies with the dataset; it comprises nine neurons for the Extrasensory dataset and 10 neurons for the Huawei dataset, each representing the number of activity classes within these datasets. The softmax activation function is employed in the output layer to provide a probability distribution over the predicted activity classes, facilitating accurate activity classification.

#### 3.7.2. Training Process

We trained the MLP using a backpropagation algorithm with a stochastic gradient descent optimizer [99,100]. A categorical cross-entropy [101,102,103] loss function was employed, suitable for the multi-class classification challenges presented by our datasets. The key elements of our training process included:Batch Size: We processed 32 samples per batch, optimizing the computational efficiency without sacrificing the ability to learn complex patterns.Epochs: The network was trained for up to 100 epochs. To combat overfitting, we implemented early stopping, which halted training if the validation loss did not improve for 10 consecutive epochs.Validation Split: To ensure robust model evaluation and tuning, 20% of our training data were set aside as a validation set. This allowed us to monitor the model’s performance and make necessary adjustments to the hyperparameters in real-time.

#### 3.7.3. Model Application and Evaluation

Following the rigorous training phase, we applied the trained MLP model to the test sets from both the Extrasensory and Huawei datasets to critically assess their effectiveness in real-world scenarios. Our evaluation strategy was comprehensive, focusing on a range of metrics that provide the accuracy and robustness of the models.

Performance Metrics: We evaluated the model based on accuracy, precision, recall, and the F1-score [104,105]. These metrics were calculated to assess the overall effectiveness of the models in correctly classifying the activities.Confusion matrix: For each dataset, a confusion matrix was generated to visually represent the performance of the model across all activity classes. The confusion matrix [106,107] helps in identifying not only the instances of correct predictions but also the types of errors made by the model, such as false positives and false negatives. This detailed view allows us to specific activities where the model may require further tuning.ROC Curves: We also plotted receiver operating characteristic (ROC) curves for each class within the datasets. The ROC curves provide a graphical representation of the trade-off between the true positive rate and the false positive rate at various threshold settings. The area under the ROC curve (AUC) was calculated to quantify the model’s ability to discriminate between the classes under study.

## 4. Experimental Setup

Evaluation of the proposed system was performed on a benchmark dataset: the Extrasensory dataset and Sussex Huawei locomotion (SHL) datasets. The experiment was performed on a Mac 2017 core i5 with 16 GB of RAM, a 3.2 GHz processor, and 512 GB of SSD.

### 4.1. Datasets Descriptions

In this section, we delve into the specifics of each dataset, highlighting their diversity and how they reflect real-world scenarios.

#### 4.1.1. The Extrasensory Dataset

The Extrasensory dataset was compiled through the utilization of a variety of sensors, including inertial, GPS, compass, and audio sensors. The data collection process was facilitated by an extra-sensory smartphone app, which aimed to monitor human physical and locomotion activities. The dataset comprises information derived from 36 individual users, with each user contributing a substantial number of instances. Data were collected through both Android and iPhone smartphones, and the dataset includes a comprehensive set of 116 labels for user-reported activities. The details of the dataset are also given in Table 1.

#### 4.1.2. The Sussex Huawei Dataset (SHL)

The Sussex Huawei Locomotion (SHL) dataset [108] is a comprehensive collection of data designed to support research in mobile sensing, particularly for the recognition of human activities and modes of transportation. It was created through a collaboration between the University of Sussex and Huawei Technologies Co., Ltd. The dataset consists of recordings from smartphone sensors, such as accelerometers, gyroscopes, magnetometers, and barometers. These sensors capture movements and environmental characteristics as people go about various activities, including walking, running, cycling, and traveling by car, bus, or train. Participants carried smartphones equipped with these sensors went through a series of movements in real-world settings, ensuring that the data was as realistic and varied as possible.

### 4.2. First Experiment: Confusion Matrix

We perform the activity classification using MLP. To evaluate the performance, we plotted the confusion matrix. In simple words, a confusion matrix is a table used for classification problems. It is used to see where the model made an error. The confusion matrices calculated for physical and localization activity for both datasets are shown in Table 2, Table 3, Table 4 and Table 5.

### 4.3. Second Experiment: Precision, Recall, and F1-Score

In this experiment, we evaluated our system by plotting precision, recall, and f1-score for individual activity. In Table 6 and Table 7, the evaluation for physical and localization activity can be seen.

### 4.4. Third Experiment: Receiver Operating Characteristics (ROC Curve)

To further assess the performance and robustness of our system, we employed the ROC curve, a well-established graphical tool that illustrates the diagnostic ability of a classification system. The ROC curve visualizes the trade-offs between the true positive rate (sensitivity) and false positive rate (1-specificity) across various threshold settings. The area under the ROC curve (AUC) serves as a single scalar value summarizing the overall performance of the classifier. A model with perfect discriminatory power has an AUC of 1, while a model with no discriminatory power (akin to random guessing) has an AUC of 0.5. In Figure 10 and Figure 11 the Roc curve is plotted.

### 4.5. Fourth Experiment: Comparison with Other Techniques

In the last experiment, the proposed system is compared with the state-of-the-art techniques. Table 8 shows the comparison of the proposed model with other state of the art techniques.

## 5. Computational Analysis 

The comparative analysis of time consumption and memory usage between the Extrasensory and Huawei datasets reveals significant differences in efficiency and resource demands. These disparities suggest diverse applicability in real-world scenarios. Specifically, the extrasensory dataset, with its higher time and memory requirements, is best suited for environments where detailed and complex activity recognition is crucial, and computational resources are less constrained, such as in clinical or controlled research settings. On the other hand, the Huawei dataset, with its lower resource demands, demonstrates suitability for consumer electronics and real-time applications, such as smartphones and wearable devices that require efficient processing capabilities. The findings show that, while the system exhibits robust performance, its deployment in resource-limited environments such as low-end smartphones or IoT devices might be challenging. Thus, our system is ideal for scenarios where precision and detailed activity recognition outweigh the need for low resource consumption, and less so for applications requiring minimal power usage and rapid processing. Figure 12 shows the analysis visually.

## 6. Discussion and Limitations

Our research has successfully demonstrated the utilization of smartphone and smartwatch sensors to accurately identify human movements and locations. By methodically cleaning, segmenting, and extracting features from raw sensor data, and employing a multi-layer perceptron for classification, our system achieved high accuracy rates. Specifically, we observed success rates of 96% and 94% for identifying physical activities over the extrasensory and SHL datasets, respectively, and 94% (Extrasensory) and 91% (SHL) for localization activities. These results represent a significant improvement over many existing methods and underscore the potential of our approach in applications where precise activity recognition is crucial.

Detailed Analysis of Key Findings

The high accuracy rates in physical activity recognition demonstrate the efficacy of the proposed system’s feature extraction and machine learning workflow. For localization activities, although slightly lower, the success rates are still competitive, emphasizing our system’s capability in varied contexts. These findings suggest that our approach could be particularly beneficial in health monitoring, urban navigation, and other IoT applications that demand reliable human activity and location data.

While our proposed system offers a promising approach for biosensor-driven IoT wearables in human motion tracking and localization, we recognize several inherent challenges that could impact its broader application and effectiveness. These limitations, if not addressed, may curtail the system’s reliability and versatility in diverse environments:GPS limitations: The GPS technology we utilize, while generally effective, can suffer from significant inaccuracies in environments such as urban canyons or indoors due to signal blockage and multipath interference. These environmental constraints can affect the system’s ability to precisely track and localize activities, particularly in complex urban settings.Data diversity and completeness: The dataset employed for training our system, though extensive, does not encompass the entire spectrum of human activities, particularly those that are irregular or occur less frequently. This limitation could reduce the model’s ability to generalize to activities not represented in the training phase, potentially impacting its applicability in varied real-world scenarios.Performance across different hardware: Our system was primarily tested and optimized on a specific computational setup. When considering deployment across diverse real-world devices such as smartphones, smartwatches, or other IoT wearables, variations in processing power, storage capacity, and sensor accuracy must be addressed. The heterogeneity of these devices could result in inconsistent performance, with higher-end devices potentially delivering more accurate results than lower-end counterparts.Scalability and real-time processing: Scaling our system to handle real-time data processing across multiple devices simultaneously presents another significant challenge. The computational demands of processing large volumes of sensor data in real time necessitate not only robust algorithms but also hardware capable of efficiently supporting these operations.Privacy and security concerns: As with any system handling sensitive personal data, ensuring privacy and security is paramount. Our current model must incorporate more advanced encryption methods and privacy-preserving techniques to safeguard user data against potential breaches or unauthorized access.

## 7. Conclusions and Future Work

In this study, we successfully developed a comprehensive system capable of effectively recognizing human physical activities and localization through a combination of inertial and GPS sensor data. Our system initiates with denoising the raw signals using Butterworth and median filters to reduce noise while preserving essential signal characteristics. This is followed by the Hamming windowing technique and segmentation processes that structure the data for more effective analysis. Subsequently, we extract and optimize statistical features using the variance threshold selection method and Yeo–Johnson power transformation, respectively, significantly enhancing the relevance and performance of these features in the activity classification process. The final classification of activities is executed through a multilayer perceptron (MLP), which provides a robust model capable of predicting various types of human movements and positions. The findings from our research offer significant implications for the development of smarter, more responsive wearable and mobile technology. By showcasing high accuracy in activity recognition, our system lays a foundation for improved user interaction and monitoring across various applications, spanning from personal fitness tracking to patient health monitoring in medical settings. The successful integration of sensor data for precise activity and location recognition paves the way for more intuitive and context-aware devices.

Moving forward, several enhancements and extensions are proposed to further enrich the capabilities of our system and its applicability to a broader range of real-world scenarios. First, integrating additional types of sensor data, such as environmental and biometric sensors, could provide a more complex understanding of the context and improve the accuracy and reliability of activity recognition. Second, developing adaptive algorithms that can dynamically adjust to changes in the environment or user behavior would make the system more responsive and versatile. Additionally, scalability improvements are crucial, and future work will focus on optimizing the system to more efficiently handle larger, more diverse datasets. This will involve refining our algorithms to manage increased computational demands while enhancing performance. Another important direction for future research involves enhancing the real-time processing capabilities of our system, which is essential for applications requiring immediate responses, such as emergency services or live health monitoring. Furthermore, given the sensitive nature of the data involved in our system, advancing data privacy and security measures will be a priority. We plan to explore sophisticated encryption methods and privacy-preserving data analytics to ensure the security and privacy of user data.

## Figures and Tables

**Figure 1 sensors-24-03032-f001:**
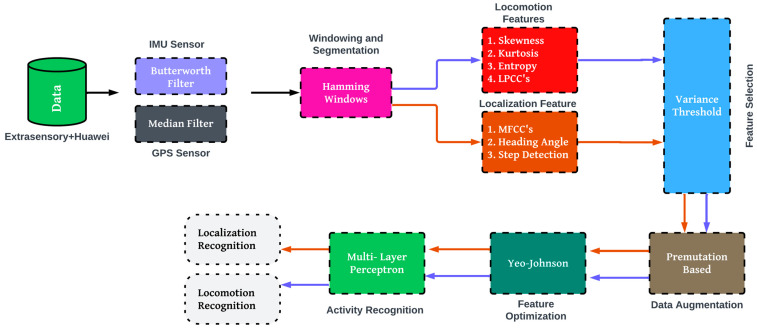
The proposed system architecture.

**Figure 2 sensors-24-03032-f002:**
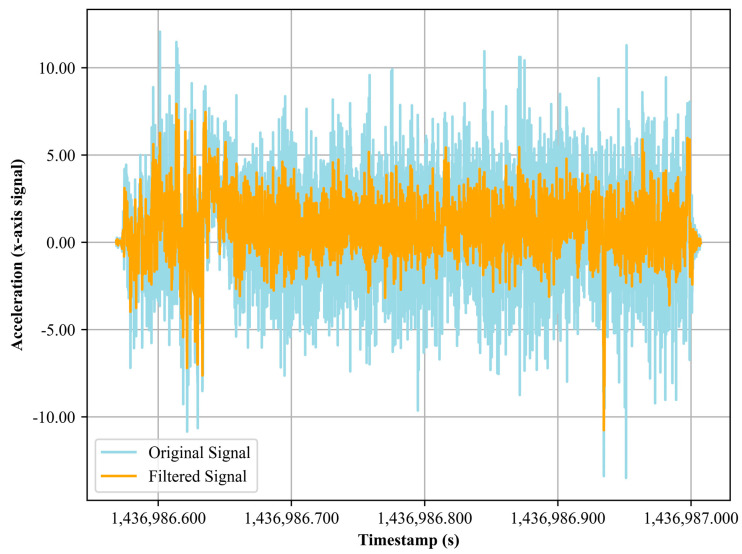
The accelerometer x axis noisy vs. filtered signal.

**Figure 3 sensors-24-03032-f003:**
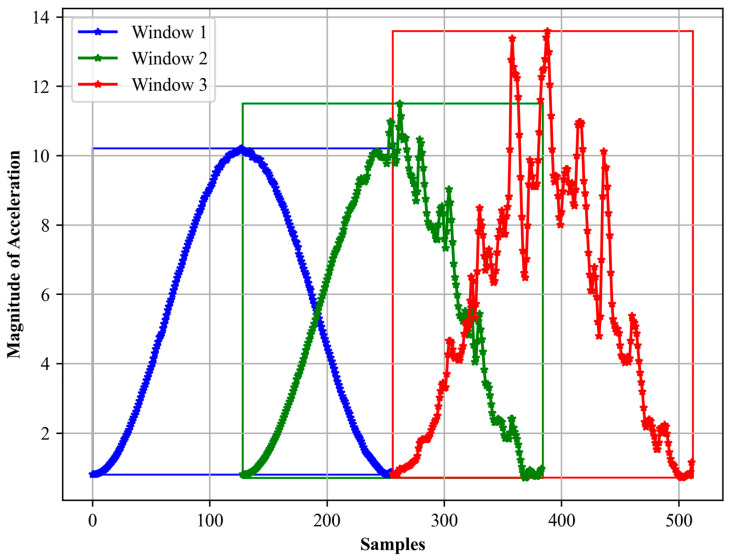
Hamming windows first 3 windows for accelerometer data.

**Figure 4 sensors-24-03032-f004:**
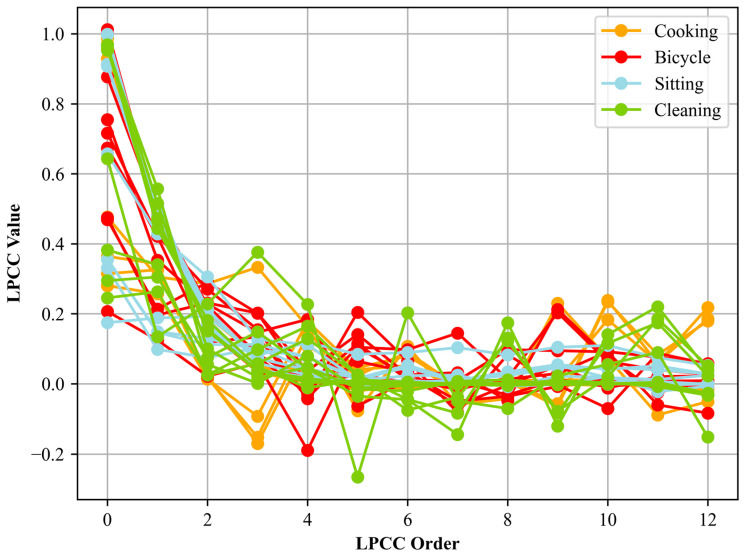
LPCCs are calculated for different activities.

**Figure 5 sensors-24-03032-f005:**
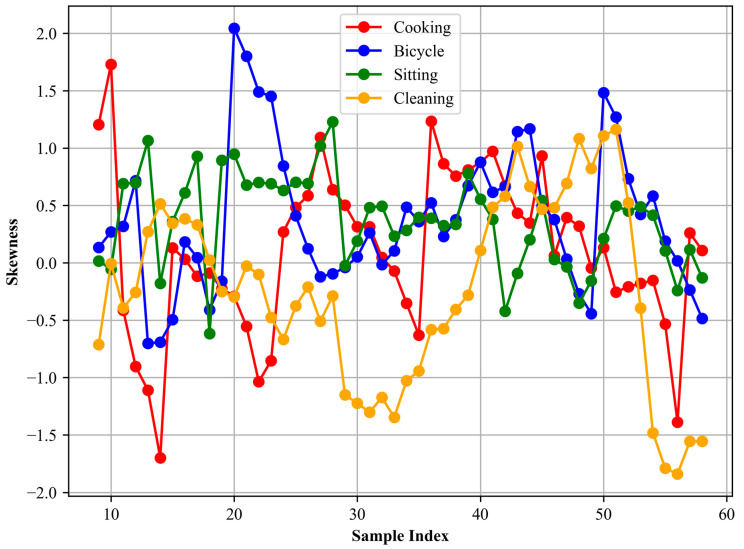
Skewness is calculated for different activities.

**Figure 6 sensors-24-03032-f006:**
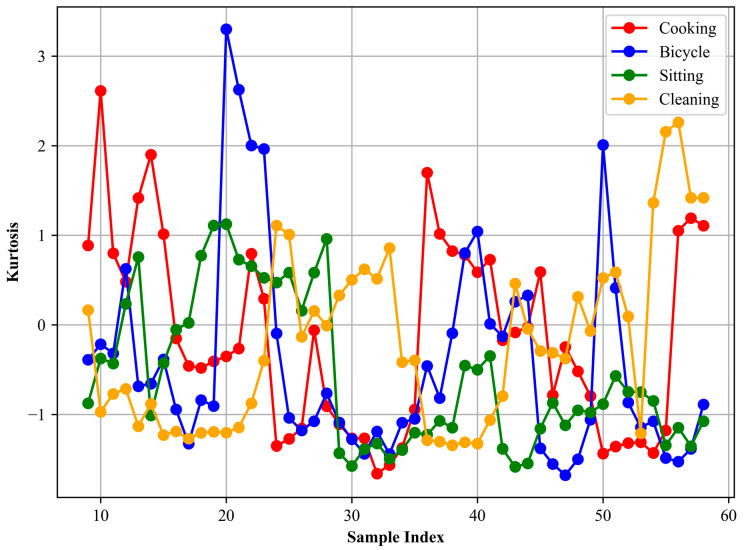
Kurtosis is calculated for different activities.

**Figure 7 sensors-24-03032-f007:**
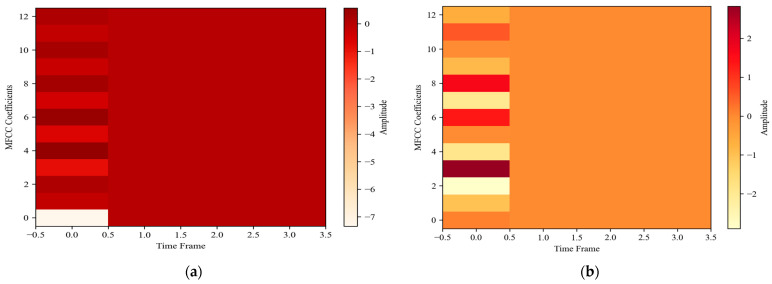
MFCCs are calculated for (**a**) indoor and (**b**) outdoor activity.

**Figure 8 sensors-24-03032-f008:**
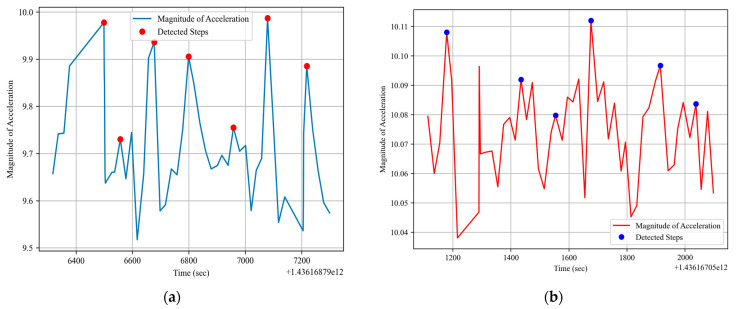
Steps detected for (**a**) indoor and (**b**) outdoor activity.

**Figure 9 sensors-24-03032-f009:**
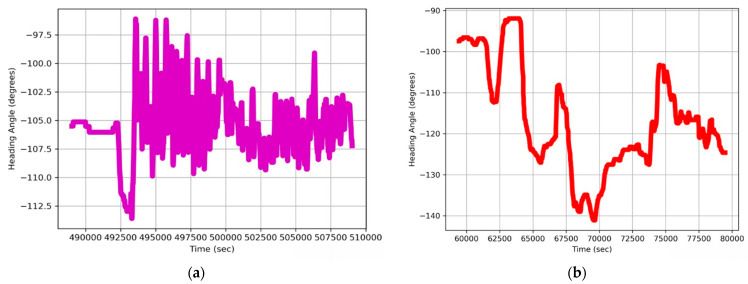
Heading angle calculated for (**a**) indoor and (**b**) outdoor activity.

**Figure 10 sensors-24-03032-f010:**
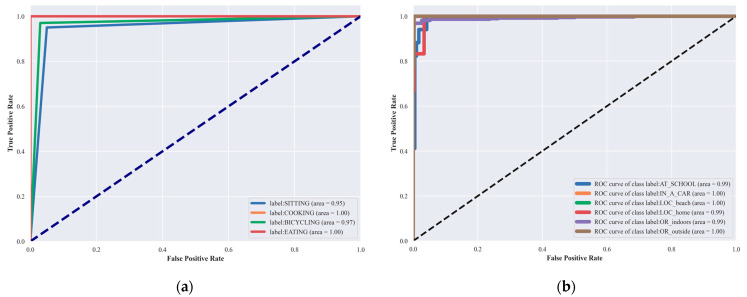
ROC curves: (**a**) physical and (**b**) localization activity over extrasensory dataset.

**Figure 11 sensors-24-03032-f011:**
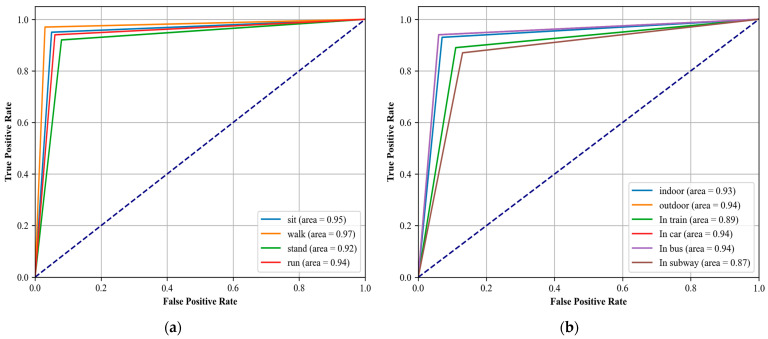
ROC curves: (**a**) physical and (**b**) localization activity over the SHL dataset.

**Figure 12 sensors-24-03032-f012:**
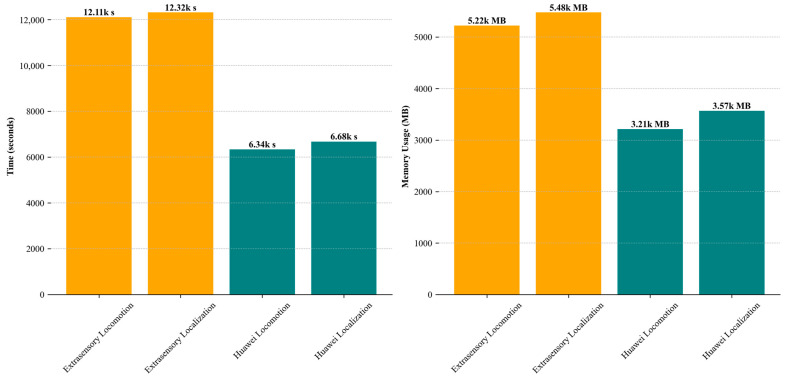
Time and memory usage analysis of the proposed system.

**Table 1 sensors-24-03032-t001:** Description of the extrasensory dataset.

Sensors	Signal Type	Sampling Rate (Hz)	Duration (sec)	Number of Recordings
Accelerometer	Acceleration	32	2	308,306
Gyroscope	Angular Velocity	32	2	291,883
Magnetometer	Magnetic Field	32	2	282,527
Location	Latitude, Longitude	1	2	273,737

**Table 2 sensors-24-03032-t002:** Confusion matrix over the Extrasensory dataset for physical activity.

Obj. Classes	Sitting	Eating	Cooking	Bicycle
**sitting**	**0.95**	0.01	0.03	0.00
**eating**	0.00	**1.00**	0.00	0.00
**cooking**	0.00	0.00	**1.00**	0.00
**bicycle**	0.03	0.00	0.00	**0.97**
		**Mean Accuracy=**	**96.61%**	

**Table 3 sensors-24-03032-t003:** Confusion matrix over the Extrasensory dataset for localization activity.

Obj. Classes	Indoors	Outdoors	Home	School	Car
**Indoors**	**1.00**	0.00	0.00	0.00	0.00
**Outdoors**	0.00	**1.00**	0.00	0.00	0.00
**Home**	0.05	0.06	**0.80**	0.02	0.07
**School**	0.02	0.02	0.03	**0.90**	0.03
**Car**	0.00	0.00	0.00	0.00	**1.00**
		**Mean Accuracy=**	**94.28%**		

**Table 4 sensors-24-03032-t004:** Confusion matrix over the SHL dataset for physical activity.

Obj. Classes	Sit	Walk	Stand	Run
**Sit**	**0.96**	0.00	0.04	0.00
**Walk**	0.03	**0.97**	0.00	0.00
**Stand**	0.03	0.03	**0.92**	0.02
**Run**	0.02	0.01	0.03	**0.94**
		**Mean Accuracy=**	**94.75%**	

**Table 5 sensors-24-03032-t005:** Confusion matrix over the SHL dataset for localization activity.

Obj. Classes	Indoor	Outdoor	In Train	In Car	In Bus	In Subway
**Indoor**	**0.93**	0.00	0.05	0.02	0.00	0.00
**Outdoor**	0.00	**0.95**	0.04	0.00	0.00	0.01
**In train**	0.01	0.03	**0.89**	0.02	0.05	0.00
**In car**	0.00	0.01	0.01	**0.94**	0.00	0.04
**In bus**	0.03	0.02	0.07	0.00	**0.88**	0.00
**In subway**	0.03	0.00	0.03	0.00	0.02	**0.92**
		**Mean Accuracy**	**=91.83%**			

**Table 6 sensors-24-03032-t006:** Precision, recall, and F1-score over physical activity.

Classes	Extrasensory			SHL		
Activities	Precision	Recall	F1-Score	Precision	Recall	F1-Score
Sitting	0.95	1.00	0.92	-	-	-
Eating	1.00	0.80	0.90	-	-	-
Cooking	1.00	0.89	0.95	-	-	-
Bicycle	0.97	0.95	0.96	-	-	-
Sit	-	-	-	0.92	0.96	0.94
Stand	-	-	-	0.94	0.92	0.93
Walking	-	-	-	0.96	0.97	0.97
Run	-	-	-	0.95	0.94	0.92

**Table 7 sensors-24-03032-t007:** Precision, recall, and F1-score over localization activity.

Classes	Extrasensory			SHL		
Activities	Precision	Recall	F1-Score	Precision	Recall	F1-Score
Indoors	1.00	0.94	0.91	-	-	-
Outdoors	1.00	1.00	0.95	-	-	-
School	0.84	1.00	0.92	-	-	-
Home	0.90	0.85	0.88	-	-	-
Car	1.00	1.00	1.00	-	-	-
Indoor	-	-	-	0.93	0.93	0.93
Outdoor	-	-	-	0.94	0.95	0.94
In train	-	-	-	0.82	0.89	0.85
In car	-	-	-	0.96	0.94	0.95
In subway	-	-	-	0.95	0.92	0.93
In bus	-	-	-	0.93	0.88	0.90

**Table 8 sensors-24-03032-t008:** Comparison of proposed MLP with other methods.

Method	Accuracy %	
	Extrasensory	SHL
Vaizman et al. [109]	0.83	-
Vaizman et al. [98]	0.83	-
Asim et al. [108]	0.87	-
Sharma et al. [110]	-	0.92
Akbari et al. [111]	-	0.92
Brimacombe et al. [112]	-	0.79
**Proposed**	**0.94**	**0.91**

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
