# Peer review of "Biosensor-Driven IoT Wearables for Accurate Body Motion Tracking and Localization"

_sensors, 2024, doi:10.3390/s24103032_

Round 1

Reviewer 1 Report

Comments and Suggestions for Authors

Comments on the Quality of English Language

English needs to be further improved.

Author Response

Thanks for your valuable comments. 

All comments are answered one-by-one in attached file. 

Please consider.

Reviewer 2 Report

Comments and Suggestions for Authors

Comments on the Quality of English Language

Moderate editing of English language required

Author Response

(The authors gave the same response as above.)

Reviewer 3 Report

Comments and Suggestions for Authors

The article delves into the burgeoning field of human locomotion identification via smartphone sensors, highlighting its potential across various sectors. While existing research has mainly focused on locomotion activities, this study introduces a system that also recognizes human localization patterns. 

The integration of both inertial and GPS sensors to recognize both physical activities and localization patterns is a novel contribution. This comprehensive approach expands the applicability of the system across various domains, enhancing its potential impact.

The paper employs a robust methodology, including preprocessing techniques like Butterworth and median filters to denoise raw sensor data, as well as advanced feature extraction and selection methods. The use of Hamming windowing for segmentation and the Variance threshold feature selection method ensures the extraction of relevant information from the sensor data, leading to accurate classification results.

The thorough evaluation using K-fold cross-validation on two distinct datasets, Extrasensory and Sussex Huawei Locomotion (SHL), strengthens the reliability of the proposed system's performance. By testing the system on diverse datasets, the study demonstrates its adaptability and effectiveness across different scenarios.

 on the other side, while the paper claims to outperform previous state-of-the-art methods, it lacks a detailed comparative analysis with existing approaches.  Including comparisons with baseline methods or competing approaches would provide valuable insights into the system's performance relative to existing solutions.

The paper does not discuss potential challenges encountered during the implementation of the proposed system. Factors such as computational complexity, resource constraints on mobile devices, or real-world deployment issues could impact the practical feasibility and scalability of the system. Addressing these challenges and providing insights into how they were mitigated would enhance the practical relevance of the study.

 Acknowledging the limitations of the proposed system, such as its performance in specific environments or its scalability to large-scale deployments, would provide valuable context for readers and guide future research directions. Additionally, discussing potential extensions or enhancements to the system could further enrich the contribution of the study.

While the paper presents a promising approach for biosensor-driven IoT wearables, addressing the identified weaknesses would strengthen its impact and relevance in the field of human motion tracking and localization.

Author Response

(The authors gave the same response as above.)

Reviewer 4 Report

Comments and Suggestions for Authors

Presented paper discuss very active developing field of research, related with using smartphones data to track body motion and localization. Basic idea presented in the paper looks perspective and interesting but authors does not include any information concerning main component of the system – multy-layered perceptron. Also paper have many issues and require very strong major revision:

1.      Line 97, please expand abbreviations ECG and EMG. Also please check all text, because there are a lot of abbreviations that are not expanded.

2.      Lines 101-103 contain the same information as lines 51-53, please reconsider text.

3.      Lines 165-166 there are typos in formula 1 notation.

4.      Section 3; please describe in details pouch how you came to presented system methodology.

5.      Formula 4 line 223 – there are typos in formula 4;

6.      Figure 5 and 6, please make colors of lines corresponding, e.g. the same colour for the same activity in all figures.

7.      There is no and information concerning used multy-layer perceptron, results of its training and application.

8.      Discussion section should be expand and discuss main results of research.

9.      Conclusion should be expanded and show perspectives of results.

1  Please provide editing of all paper with respect to requirements and guidelines of MDPI Sensors journal.

Author Response

(The authors gave the same response as above.)

Round 2

Reviewer 1 Report

Comments and Suggestions for Authors

So the opinion writers have revised it. The authors are advised to improve the quality of English.

Comments on the Quality of English Language

To be improved.

Author Response

Thanks for your valuable comments. 

Answers are given in attached file. 

Please consider.

Reviewer 2 Report

Comments and Suggestions for Authors

The changes made in this article are good, but the full text needs to be checked for syntax errors.

Comments on the Quality of English Language

Minor editing of English language required

Author Response

(The authors gave the same response as above.)

Reviewer 3 Report

Comments and Suggestions for Authors

The majority of our remarks have been addressed and revised. Consequently, I recommend accepting the paper.

Author Response

(The authors gave the same response as above.)

Reviewer 4 Report

Comments and Suggestions for Authors

Paper can be accepted for publication in present form. 

Author Response

(The authors gave the same response as above.)
